# Nutritional Value of Banded Cricket and Mealworm Larvae

**DOI:** 10.3390/foods12224174

**Published:** 2023-11-20

**Authors:** Jagoda Kępińska-Pacelik, Wioletta Biel, Cezary Podsiadło, Grzegorz Tokarczyk, Patrycja Biernacka, Grzegorz Bienkiewicz

**Affiliations:** 1Department of Monogastric Animal Sciences, Division of Animal Nutrition and Food, West Pomeranian University of Technology in Szczecin, Klemensa Janickiego 29, 71-270 Szczecin, Poland; jagoda.kepinska-pacelik@zut.edu.pl; 2Department of Agroengineering, Division of Irrigation, West Pomeranian University of Technology in Szczecin, Juliusza Słowackiego 17, 71-434 Szczecin, Poland; cezary.podsiadlo@zut.edu.pl; 3Department of Fish, Plant and Gastronomy Technology, Faculty of Food Sciences and Fisheries, West Pomeranian University of Technology in Szczecin, Papieża Pawła VI 4, 71-459 Szczecin, Poland; grzegorz.tokarczyk@zut.edu.pl (G.T.); patrycja.biernacka@zut.edu.pl (P.B.); 4Department of Commodity Science, Quality Assessment, Process Engineering and Human Nutrition, Faculty of Food Sciences and Fisheries, West Pomeranian University of Technology in Szczecin, Papieża Pawła VI 4, 71-459 Szczecin, Poland; grzegorz.bienkiewicz@zut.edu.pl

**Keywords:** edible insects, fatty acids, minerals, proximate composition

## Abstract

Insect farming is more ecological than traditional animal farming, as it requires less water and contributes to lower greenhouse gas emissions. In our study, banded cricket (BC) and mealworm larvae (ML) were analyzed. The proximate composition was determined according to Association of Official Analytical Chemists. The mineral content was determined by colorimetry and mass spectrometry. Fatty acid methyl esters (FAMEs) were obtained from the samples and separated using a gas chromatography apparatus, coupled with a mass spectrometer. Our research confirmed that insects are a rich source of protein, with ML containing significantly more protein than BC (74.41 and 65.66 g/100 g dry matter (DM), respectively). In terms of the content of macrominerals, ML was significantly richer than BC, especially in terms of magnesium content (8.75 g/100 g DM). In terms of the content of saturated fatty acids, BC contained almost twice as much as ML (40.05 and 24.74% of the sum of fatty acids, respectively). EPA and DHA were only detected in the fat of BC. The presented results prove that both ML and BC can be good sources of protein both in human and companion animal diets. The component that is predominantly high in insects is fat, with a favorable fatty acid profile, especially in terms of polyunsaturated fatty acids. This study contributes new knowledge on the nutritional value of edible insects. In this research, we included three different nitrogen conversion factors for crude protein content. Our results partially confirm previous studies by other authors, although they provide new information on the content of fatty acids.

## 1. Introduction

To meet the nutritional needs of a global population of 9 billion according to current dietary patterns, overall food production is predicted to increase by 70% by 2050 [1]. This phenomenon will have serious consequences, including an additional 30% increase in global greenhouse gas emissions resulting from the breeding of animals for slaughter [2,3]. Given the environmental impacts of animal husbandry, including land use, greenhouse gas emissions, and water pollution, increasing production is not a sustainable solution to meet the nutritional needs of an ever-growing human population [4]. Therefore, currently, global organizations are calling for the search for alternative sources of protein, such as insects, and the conversion of breeding practices based on slaughtering animals into the breeding of edible insects [5]. In addition to lowering greenhouse gas emissions, insects use less water and require much less space to breed, and they can feed on some of the waste from food production. Moreover, thanks to the high efficiency of feed processing, an average of 2 kg of fodder mass can be produced from 1 kg of insect biomass [6]. Insects have been the food of humans and animals for thousands of years, especially in the southern part of the globe. It is also worth bearing in mind the fact that there are many species of edible insects specific to a given country or region. For example, weaver ants (*Occophylla smaragdina*) are widespread in the Asia–Pacific region and are found from China’s south to northern Australia and as far west as India. A total of 81 species are eaten in Arunachal Pradesh by the Galo and Nyishi tribes [7]. In southern Africa, one of the most significant edible insects seems to be the phane caterpillars of a saturniid emperor moth (*Gonimbrasia belina*), reaching a yearly trade value of more than $85 million [8].

Currently, insects are an important part of the human diet and traditional food, and can contribute to food, environmental and financial security through local business opportunities [9]. Insects have been a source of protein for thousands of years not only in the case of humans. The same is true for companion animals, whose ancestors had insects as the basis of their diet in many cases. To this day, wild relatives of dogs and cats eat insects in their natural environment [10,11]. In the case of dogs and cats, hopes are associated with insects for their use in foods for animals with food allergies [12].

The nutritional value of insects depends on many factors. Firstly, the proximate composition is known to be species-related [13]. The nutrient content of insects can also be influenced by rearing technology, the composition of the feed, their origin, and sex [14,15,16,17]. The quality of edible insect lipids is comparable to that of pork, chicken fat, and vegetable margarine [18,19]. From a nutritional perspective, certain insect species are comparable or even superior to commercial nutritional sources [20].

One of the most important factors determining the composition of edible insects seems to be their diet. It is easy to modify the composition of insects by providing them with a diet of different nutritional value. Nowadays, chicken feed and soybean meal are relatively common for insect breeding, but innovative substrates are also gaining popularity. The key aspects of optimizing the composition of insect feed include, for example, the content of protein and fat, which are the most important nutrients. An example of modifying the composition of crickets is the study by Bawa et al. [21], in which a substrate with the addition of fresh pumpkin pulp with a high carbohydrate content increased the fat content in the insects. In the same research, high protein content in feed was shown to increase not only the protein content but also the amount of sodium, calcium, phosphorus, and potassium in crickets [21].

In the case of the mealworm, it was shown that larvae fed on brewer’s grain had a higher content of crude protein and carbohydrates than larvae fed on bread or cookies [22]. Zhang et al. [23] observed that mealworms fed on fiber-rich substrates had a higher protein content and lower fat content than mealworms fed on fiber-free substrates. Interestingly, in the research of Finke [24], insects fed diets containing fish oil accumulated eicosapentaenoic acid but not docosahexaenoic acid, even though both were present in the diets. Despite the scientifically proven benefits of breeding and consuming insects, entomophagy is still a fairly common limitation. It has been shown that people who have eaten insects in the past are more likely to eat them again, while people who have never tried insects before are more likely to avoid eating them [9]. Therefore, in order to overcome the negative attitude towards direct entomophagy, especially caused by disgust reactions, it is important to make people aware of the benefits of entomophagy [25]. On the other hand, information about health, taste, and sustainability did not affect willingness to try insect food [26]. What is more, acceptance of whole insects is more influenced by species than size [27].

From 2021, EU law has allowed insect food to be sold in stores under certain conditions. Such a product must be considered as “novel-food” by obtaining the appropriate authorization from the European Food Safety Authority (EFSA) or its manufacturer must prove that it is a product commonly consumed in the European Union. The term “livestock” is most often associated with cattle, pigs, and poultry. Recently, species that can be used as raw material for the production of feed for certain animals in the European Union have also been added to this list. Pursuant to Regulation 893 of the Commission (EU) from 2017 [28], the number of livestock species has been extended to include the following insect species in connection with the provisions on processed animal protein:

(1) flies: black fly (*Hermetia illucens*) and housefly (*Musca domestica*);

(2) beetles: mealworm (*Tenebrio molitor*) and glossy mildew (*Alphitobius diaperinus*);

(3) crickets: house cricket (*Acheta domesticus*), banded cricket (*Gryllodes sigillatus*), and Cuban cricket (*Gryllus assimilis*).

This regulation allows the use of processed protein from insects for the production of feed for aquaculture animals, fur animals, and pet food [28]. In 2021, the use of insects as processed animal protein in feed for poultry and pigs was allowed [29]. In human nutrition, legal regulations are a key condition for the development of insect breeding and effective marketing of food with their participation. The above-mentioned species do not carry pathogens characteristic of plants, animals, or humans; they are not invasive; they do not cause diseases in humans and animals; they do not adversely affect crops; and they are not protected. According to the scientific opinion of the EFSA as of 13 January 2021 [30], mealworm larvae are safe for human consumption. Previously approved were the locust and house cricket [31]. In 2023, the EU approved the glossy mildew [32]. It should also be noted that not only whole insects but also their parts and products based on them (such as insect meal) can be placed on the EU market [33].

Despite the many benefits of replacing slaughtered animals with insects, there are few recent literature reports comparing selected species of edible insects. Therefore, the aim of this study was to evaluate the nutritional value of the subimago banded cricket (*Gryllodes sigillatus*) (BC) and mealworm larvae (*Tenebrio molitor*) (ML), taking into account the proximate composition, content of minerals, and fatty acids.

## 2. Materials and Methods

### 2.1. Material

Insects of two species (subimago banded cricket (*Gryllodes sigillatus*) and mealworm larvae (*Tenebrio molitor*)) were purchased from one insect farm located in Poland. Both species were in the subimago stage—before reaching adulthood. Both insect species were fed the same feed (analytical composition: crude protein 16%, crude fat 3%, crude ash 8%, crude fiber 6%, vitamin A 8500 IU/kg, vitamin D3 1100 IU/kg, vitamin E 60 mg/kg). Insects were anesthetized using a low temperature in a freezer for two days. After removing the insects from the freezer, they were dried in a laboratory oven (type UM300, Memmert, Schwabach, Germany) at 60 °C for 48 h in order to obtain an air-dry mass. The insects were then ground in a laboratory mill, type KNIFETEC 1095 (Foss Tecator, Höganäs, Sweden), and placed in sterile containers. They were stored at a cooling temperature until analyses.

### 2.2. Proximate Composition

Dry matter, crude protein, crude fat, crude fiber, crude ash, and total carbohydrates were determined according to the Association of Official Agricultural Chemists [34]. The analyses were performed in three repetitions, and the tables with the results contain mean values. Chemical reagents for the analyses were obtained from Avantor Performance Materials Poland S.A. To determine the dry matter, the samples were dried at 105 °C until constant weight in the laboratory oven (method 945.15) (type UM300, Memmert, Schwabach, Germany). Crude fat was determined by the Soxhlet method with diethyl ether as solvent (method 945.16); crude ash by burning in a muffle furnace at 580 °C for 8 h (method 920.153); crude protein content was determined by three different nitrogen conversion factors (4.76, 5.60, 6.25 × total nitrogen), measured by the Kjeldahl method using a Büchi Scrubber B-414 digestion apparatus and a Büchi 324 distillation set (Büchi Labortechnik AG, Flawil, Switzerland) (method 945.18); the crude fiber was determined with an ANKOM^220^ analyzer (ANKOM Technology, New York, NY, USA). Total carbohydrates (TC) were calculated according to Equation (1) [34,35].
TC (%) = 100 − (% moisture + % crude protein + % crude fat + % crude ash + % crude fiber)(1)

### 2.3. Minerals

The total content of potassium (K), calcium (Ca), magnesium (Mg), iron (Fe), manganese (Mn), zinc (Zn), copper (Cu), cadmium (Cd), cobalt (Co), chromium (Cr), nickel (Ni), and lead (Pb) was determined by wet mineralization in a mixture of nitric acid (V) and perchloric acid (VII) [36]. The analyses were performed with an atomic absorption spectrometer (Thermo Fisher Scientific iCE 3000 Series, Waltham, MA, USA). For the determination of Ca, K, Mg, and Na the wavelengths were determined: K: 766.5 nm; Ca: 422.7 nm; Mg: 285.2 nm; Na: 589.0 nm. For the determination of Fe, Mn, Zn, Cu, Cd, Co, Cr, Ni, and Pb, the wavelengths were as follows: Fe: 248.3 nm; Mn: 279.5 nm; Zn: 213.9 nm; Cu: 324.8 nm; Cd: 228.8 nm; Co: 240.7 nm; Cr: 357.9 nm; Ni: 232.0 nm; Pb: 207.2 nm. The calculation of the content of the individual elements was started with a standard curve taking into account the weight of the test portion and the dilutions used. The phosphorus content was determined by the Egner-Riehm colorimetric method, with ammonium molybdate [37] on a Specol 221 apparatus spectrophotometer (Carl Zeiss, Jena, Germany). The material for P concentration analyses was subjected to mineralization in concentrated sulfuric acid (H_2_SO_4_) and perchloric acid (HClO_4_). The absorbance value of the sample, determined spectrophotometrically, from P_2_O_5_ to total phosphorus, was calculated according to the chemical equivalent (0.436). The analyses were performed in three repetitions, and the tables with the results contain mean values. The credibility of the method has been confirmed by comparative studies, including the calibration curve, using the calibration series method. The accuracy of the analytical methods was verified based on certified reference material skimmed milk powder (ERM^®^-BD151, European Commission’s Joint Research Centre, Geel, Belgium), which was obtained from the Institute for Reference Materials and Measurements (IRMM, Geel, Belgium). The results of analyzing the nutrients in the quality control sample IAEA V-10 are given in Appendix A. As shown in Appendix A, the results passed the criteria [38]. Therefore, the methods that were used are reliable for analyzing insects.

### 2.4. Fatty Acids (FAs)

Fatty acid methyl esters (FAMEs) were obtained from the samples by alkaline hydrolysis of the extract of lipids [39] with 0.5 N sodium methylate (CH_3_ONa) [40]. Next, the FAMEs were separated using a gas chromatography apparatus, coupled with a mass spectrometer (Agilent Technologies 7890A, Santa Clara, CA, USA), and equipped with a split/splitless type injector. The separation conditions of FAME were as follows: SPTM 2560 column, length: 100 m, 0.25 mm ID, and 0.20 lm film, catalog no. 24056; helium as carrier gas at a constant flow rate of 1.2 mL/min; split 1:50; temperature of injector: 220 °C; temperature of detector: 220 °C; programmed temperature of furnace: 140 °C (5 min) increased to 240 °C at a pace of 4 °C/min; time of analysis: 45 min [41]. The qualitative interpretation of the chromatograms was based on the comparison of retention times and mass spectra of the particular FAMEs of the sample with those of the analogous FAME standards by Sigma Company, Tokyo, Japan (Lipid Standard). The analyses were performed in three repetitions, and the tables with the results contain the mean values. As an internal standard, C19:0 (CAS 646-30-0, Merck, Warsaw, Poland) was used.

### 2.5. Index of Atherogenicity (IA)

The atherogenic potential of the FAs was determined using the index of atherogenicity (IA) [42], according to the following equation:(2)$$IA=\frac{C12:0+\left(4\times C14:0\right)+C16:0}{\Sigma UFA}$$

### 2.6. Index of Thrombogenicity (IT)

The thrombogenic potential of FAs was determined using the index of thrombogenicity (IT) [42], according to the following equation:(3)$$IT=\frac{C14:0+C16:0+C18:0}{\left(0.5\times \Sigma MUFA\right)+\left(0.5\times \Sigma n-6PUFA\right)+\left(3\times \Sigma n-3PUFA\right)+(n-3/n-6)}$$

### 2.7. Statistical Analyses

Statistical analyses of the results were performed using the Statistica 13.1 program, with a significance level of *p* = 0.05, using the post hoc test of Tukey’s Significant Difference [43].

## 3. Results

In our research, we analyzed the proximate composition (Table 1), the content of minerals (Table 2 and Table 3), and the fatty acids (Table 4). A significantly higher protein content was found in ML: 74.41 g/100 g DM compared to 65.66 g/100 g DM, which was found in BC. In the case of fat, a higher level was found in BC, which contained almost three times as much fat as ML (21.51 g/100 g DM and 9.27 g/100 g DM, respectively). However, significantly higher levels of crude fiber and crude ash were found in ML. In the case of total carbohydrates, no significant differences were observed between ML and BC.

Of the macrominerals, only the content of sodium was higher in BC than in ML (0.42 and 0.37 g/100 g DM, respectively). Compared to the banded cricket, mealworm larvae are a more concentrated source of macrominerals, being richer in Ca, P, K, and Mg. In the case of Mg, the amounts found in ML were almost eight times higher than in BC (8.75 and 1.06 g/100 g DM, respectively). In the case of Ca, a higher level was found in ML compared to BC (0.24 g/100 g DM and 0.14 g/100 g DM). Higher phosphorus content was also found in ML (1.24 g/100 g DM). The potassium content was also found to be higher in ML (1.54 g/100 g DM).

In the case of trace elements, the opposite situation can be observed: BC is a better source compared to ML. The amounts of trace elements found in BC were significantly higher than in ML. Particularly noteworthy is the level of Mn, the amounts of which found in BC were more than twice as high as in ML (1.89 and 0.87 mg/100 g DM, respectively). The levels of analyzed heavy metals (Cd, Co, Cr, Ni, Pb) were below the limit of detection.

Fat content in BC (21.51%) was twice as high as in ML (9.27%) (Table 1). In the fat of the tested insects, 15 and 12 fatty acids were determined, respectively. Total fatty acids were 655 mg/g in BC and 792 mg/g in ML. Banded crickets had a twice as high percentage of SFAs compared to mealworms, while in the case of MUFAs, this relationship was the opposite. On the other hand, the percentage of PUFA in both species was comparable and amounted to 33.82% for BC and 35.09% for ML. In the group of saturated acids, palmitic acid was the most abundant, and its content was higher in BC compared to ML (153 and 123 mg/g of fat). In the group of monounsaturated acids, the highest percentages were found in C18:1 n-9 acid. Comparing the absolute values of this acid between the tested samples, it was found that ML lipids had twice as much of this acid as BC lipids (Table 4). The greatest differences in the composition of fatty acids were found in the group of polyunsaturated acids. The BC lipids had a total of five PUFAs and the ML lipids only two. The BC lipids were dominated by C18:2 n-6 acid in the amount of 200 mg/g of fat, and in the ML lipids, it was 269 mg/g of fat. The BC lipids also contained EPA and DHA, which were not found in the ML lipids. Comparing the ratio of n-6 to n-3 acids, it was found that, in terms of nutrition, BC lipids have a better ratio, as this ratio is almost two times lower compared to ML (17 for BC and 33 for ML, respectively).

The AI index was 0.29 for ML, while for BC it was half as high and amounted to 0.43 (Table 4). The IT index in BC was 1.04, while in ML 4.65.

## 4. Discussion

Environmental aspects and the depletion of land resources lead to the search for protein sources alternative to slaughtered animals. There is hope in insects, which have been the food of indigenous people in many regions of the world for thousands of years [12]. Also, in the case of animals, they are an important part of their diet. These facts prompt the scientific community to look more closely at insects, their chemical composition, and the presence of compounds that have a beneficial effect on the health of consumers. Protein, fat, mineral, and vitamin contents in insects generally satisfy the requirements of healthy food, although there is considerable variation associated with insect species, collection site, processing method, insect life stage, rearing technology, and insect feed [44]. Studies have shown that insect meal can partially or completely replace fishmeal (FM), which is commonly used in feed and pet food. Cricket meal (CM) (*Gryllus bimaculatus*) can improve growth performance (weight gain rate and specific growth rate), improve blood health and liver antioxidant capacity, and significantly alter the gut microbiology of animals, suggesting that FM could be replaced by CM as the source of insect protein in the diet [45]. Moreover, black soldier fly, housefly, and mealworm larvae have been found to be highly digestible and fermentable in dogs, suggesting that these species could successfully replace slaughtered animals in foods for companion animals [46].

Especially important is the content of protein, which is the most essential nutrient. The nutritional value of protein depends primarily on the raw material in which it occurs. The method of thermal processing of the raw material is also important for the structure and digestibility of the protein [47]. Protein of animal origin is important not only in the diet of humans but also in animals [48]. Therefore, new sources of protein are constantly being sought. Insects seem to be a very good source of protein; in our research, both species contained between 65.66 and 74.41 g/100 g DM of protein, which is almost twice as high as compared to the literature data on the meat of animals for slaughter [8,49,50,51]. According to Ribeiro et al. [52], banded crickets contained 65.3 g/100 g DM of protein; however, Zielińska et al. [50] determined the content of protein to be about 70 g/100 g DM. On the other hand, Gugolek et al. [53] found a protein content of 55.83 g/100 g DM in mealworm larvae. The results of Jajić et al. [54] were more similar to those found by us; the crude protein content was at the level of 38.9–71 g/100 g DM. Cricket protein concentrate (CPC) can be used as a functional ingredient in the food industry due to its excellent functional and biological properties. In addition to their high protein content and high gastrointestinal digestibility, CPCs have additional biological values, such as strong antioxidant and anti-inflammatory effects, which may be due to their high content of phenolic compounds [55]. There is some controversy in the literature regarding the protein conversion factor value used to calculate protein content based on nitrogen analysis. It is worth emphasizing that, according to Jonas-Levia and Martinez [56], the conversion factor (6.25), commonly used for protein determination, overestimates the results of protein content in insects. This method is based on estimating the total concentration of nitrogen (N) that is converted to protein by multiplying it by the nitrogen-to-protein conversion factor (N factor) for meat (6.25). According to the authors of that study, given that the insect cuticle contains large amounts of chitin, N-rich polysaccharide, and proteins tightly embedded in the matrix and is not digested by humans or domestic animals, the method overestimates the digestible protein content of insects. As indicated by Kim et al. [57], the non-protein fraction of insects, which includes nitrogen, may cause an overestimation of the protein content, and the removal of the non-protein fraction may increase the nitrogen-to-protein conversion coefficients (Kp) value of edible insects. Therefore, in our studies, the crude protein content was estimated using different Kp values, 4.76, 5.60, and 6.25, as reported by [57]. However, when discussing the results, we rely on a Kp of 6.25, which is a generally accepted conversion factor for calculating the content of animal protein.

Fats are an important nutrient because they provide flavor. The content and quality of fat, therefore, affect the palatability of food [58]. The sensory attractiveness of insect food is similar to typical food for companion animals [59], and comparing mealworm larvae and banded crickets, ML attracts more interest in male dogs, and BC in females. The high fat content in insects was also confirmed in the research by Gugołek et al. [53], where the authors found a fat level of 29.9 g/100 g DM in mealworm larvae. However, the results obtained by Jajić et al. [54] were more similar to those found by us; the crude fat content ranged from 6.1 to 45.2 g/100 g DM. In addition to the species and developmental stage, the composition of insects is determined by the feed, which is confirmed by the range of results obtained by Jajić et al. [54]. In banded crickets, the fat content was 23.5 g/100 g DM in the research of Ribeiro et al. [52], but Zielińska et al. [50] found a content of 18.23 g/100 g DM. These results confirm our research, in which the level of fat was 21.51 g/100 g DM. Both the results of our research and the cited data confirm that insects are a rich source of fat. This is important because insect fat is an excellent source of unsaturated fatty acids.

The amount and quality of fat significantly affect the energy and nutritional value of food and pet food. The fatty acids contained in fat affect the nutritional value. It is important that, apart from providing energy, fats are a source of bioactive ingredients or their precursors. In a balanced diet, polyunsaturated fatty acids are particularly important, the proper supply of which in the diet can prevent inflammation and many diseases [60]. In addition, they improve lipid metabolism, oxidative stress, and endothelial function [61]. In our research, the banded cricket was found to contain almost twice as much saturated fatty acids compared to the mealworm larvae, which is statistically significant and similar to Messina et al. [62]. SFAs are primarily a source of energy for the body [63]. In the study of Messina et al. [62], the fatty acid profile of mealworm larvae powder was characterized by an almost double predominance of monounsaturated fatty acids compared to crickets (48.06 and 24.66% of the sum of fatty acids, respectively). Similar results were obtained in our studies; however, after taking into account the fat content and comparing their percentages in absolute values, the amount of monounsaturated acid in BC was 171 mg/g of fat, and in ML 318 mg/g.

The main fatty acids in the BC and ML fats were palmitic, oleic, and linoleic acids, which together account for more than 70% of the total amount of fatty acids. These results fall within the ranges reported for these species [64,65].

Particular attention should be paid to polyunsaturated fatty acids (PUFA), which have a positive effect on the functioning of the brain and the entire nervous system, and their adequate supply is particularly important in the diet of puppies and seniors [66]. This group of fatty acids is capable of mitigating the biological processes of lipid oxidation, which result in the formation of free radicals [67]. Among the polyunsaturated acids, the most important functions of regulating pro- and anti-inflammatory processes in the body are attributed to linoleic acid (LA, C18:2 n-6) and α-linolenic acid (ALA, C18:3 n-3), which belong to essential fatty acids (EFAs). In addition to the amount of these acids in the diet, their ratio is also important. Nutritional recommendations indicate that the optimal ratio of n-6 to n-3 is 4:1 [68]. In our study, BC lipids were characterized by a better, but not optimal, ratio of these acids. Linoleic acid was present in high amounts in our research, with greater amounts found in mealworm larvae than in banded cricket (34.37 and 31.42% of fatty acids). Jayanegara et al. [69] found this fatty acid content to be 24.00 and 27.90% fatty acids in mealworm larvae and crickets, respectively.

However, these differences are not significantly different from the relationships described in the literature. It should be remembered that the composition of fatty acids, especially polyunsaturated acids, is largely determined by diet and environmental conditions [70].

The content of α-linolenic acid (ALA) in our study was at the level of 2.48 and 1.04% of the fatty acids in banded crickets and mealworm larvae, respectively, as in Jayanegara et al. [71], who found this acid content to be 1.39% of the sum of the fatty acids in crickets and 0.91% of fatty acids in mealworm larvae, respectively. Recalculating these percentages and taking into account the fat content, it turns out that 100 g of BC provides over 0.53 g of this acid and ML less than 0.1 g, and in absolute amounts, i.e., mg of fatty acid per gram of fat, BC provides twice as much as ML (15.8 and 8.15, respectively). In addition, BC fat contains small amounts of long-chain polyunsaturated acids from the n-3 family, including EPA and DHA, which are particularly important in terms of nutrition. The presence of these compounds in the diet is important because they are not synthesized by the animal’s body, so they must be supplied with the diet. In the body, ALA acid, as a result of successive processes of desaturation and elongation, is metabolized to eicosanoids, which are then used to create anti-inflammatory prostacyclins, which are attributed with a protective effect related to, among others, oxidative stress and the alleviation of inflammation [71]. The presence of long-chain fatty acids in animal fat can come directly from the feed or food they eat. Cricket lipids may contain n-3 acids with 20 and 22 carbon atoms. However, it depends on the species, method of breeding, and mainly on the feed used. In the case of the cricket, these acids can also be synthesized by the organisms of these animals. According to Tzompa-Sosa et al. [72], house cricket oil contained DHA (C22:6 n-3), suggesting elongation and desaturation of C18:3 n-3. In other studies, only EPA (C20:5 n-3), not DHA, was detected in the house cricket [18,73,74]. Blomquist et al. [75] reported the conversion of diet-derived C18:3 n-3 to 20:3 n-3, 20:4 n-3 and 20:5 n-3, as well. There is no evidence of further elongation of C20 fatty acids to C22 fatty acids in Australian field crickets, but it is possible that further elongation of EPA exists in these cricket species, but this issue requires further studies [72]. Increasing the intake of EPA and DHA can reduce the risk of coronary diseases; however, the same can also be achieved by lowering the n-6/n-3 fatty acid ratio (≤5). Some cricket species have a very high n-6/n-3 fatty acid ratio (>18/1), the same as ML, which can lead to adverse health consequences. This high n-6/n-3 ratio is due to the high content of C18:2 n-6 in the cricket feed. On the other hand, C18:2 n-6 FA is also an essential FA for humans, but its excess may adversely affect the brain [76]. According to Oonincx et al. [73], decreasing the n-6/n-3 fatty acid ratio could be achieved by including n-3 PUFAs in crickets’ diets.

No fatty acids with chains longer than 18 carbon atoms in the chain were found in oils obtained from mealworm larvae, regardless of the type of feed [77]. It can therefore be assumed that insect larvae cannot synthesize fatty acids with a chain length of more than 18 carbon atoms on their own, and they also do not absorb them in their diet. The only modification is possible only at the level of the ratio of linoleic and alpha linolenic acids, i.e., the ratio of n-6 to n-3 fatty acids.

The AI Index was 0.29 for ML, while for BC it was half as high and amounted to 0.43. Comparing these values with the values calculated for other products, it can be concluded that they are most similar to pork, for which the value of this index ranged from 0.27 to 0.31 [78], or for lamb (Iranian fat-tailed breed), where the range was from 0.53 to 0.77 [79]. The value of the IT index for BC was 1.04, which is a result comparable to that of lamb [80]. The IT index for ML was as high as 4.65 and is comparable to milk from Jersey cows, where this level ranges from 3.9813 to 4.6558, according to [81]. The IT index for ML was very unfavorable and was four times higher than for BC. Analyzing the overall health indices presented, it was found that these lipids are not beneficial in terms of health-promoting properties.

Crude fiber is the part of the dietary fiber that increases the volume of fecal masses, accelerates peristalsis, and reduces digestibility [82]. On the other hand, it gives a feeling of fullness, which can be important for dogs with excess body weight [83,84]. In our study, the mealworm larvae were richer in crude fiber compared to the banded cricket. Jajic et al. [54] found crude fiber levels of 6.3–10.4 g/100 g DM in mealworm larvae. According to research by Zielińska et al. [50] and Ribeiro et al. [52], the level of crude fiber in mealworm larvae is also very variable, although higher than in BC, and amounts to 3.65–7.1 g/100 g DM. Crude fiber includes complex carbohydrates such as cellulose, hemicellulose, and lignin that might be present in the insects’ guts [85], and therefore variability may arise depending on preparation processes, such as the degutting, cleaning and removal of parts, or the inclusion of a starvation regime prior to harvesting [86,87,88,89]. It has been shown that the chitin content determines the digestibility of crude protein in the meals of mealworm larvae. A negative and significant correlation was also detected with ADF [88]. Chitin, however, also has positive applications. As a result of chitin deacetylation, chitosan can be obtained, which due to its properties is used in the food and pharmaceutical industries [89].

While carbohydrates are the main components of most foods, animal sources are high in protein and lipids but low in carbohydrates. According to the research of Son et al. [90], the carbohydrate (TC) content in dried mealworms was 11.45 g/100 g DM, while Jajić et al. [54] found the content of TC at the level of 3.7–6.1 g/100 g DM. These results were much higher than our reported TC content of 0.62 g/100 g DM in mealworm larvae. The content of TC in the banded cricket did not differ significantly, being lower by only 0.01; it was 0.62 g/100 g DM. In the research of Zielińska et al. [50] and Ribeiro et al. [52], the content of TC in the banded cricket was found at the level of 0.1 g/100 g DM.

In our studies, mealworm larvae contained almost twice as much crude ash as the banded cricket (7.60 g/100 g DM and 4.84 g/100 g DM, respectively). Jajic et al. [54] found fluctuations in the crude ash content in mealworms: 3.48–7.5 g/100 g DM; our results were higher. In the case of the banded cricket, based on research by Zielińska et al. [50] and Ribeiro et al. [52], it can be stated that the level of crude ash in this species ranges from 4.74 to 4.2 g/100 g DM, and our results were similar to the results of these authors.

Similar to Zielińska et al. [50] and Ribeiro et al. [52], in our research, the micromineral contained in the largest amount in the banded cricket was potassium, which is of great importance for the proper functioning of the nervous and muscular systems. In mealworm larvae, the most abundant micromineral was magnesium, which is responsible for the proper functioning and transmission of signals in the nervous system. In the research by Jajić et al. [54], phosphorus and potassium were present in the greatest amount in mealworms.

In the case of the trace elements analyzed in our research, zinc was found in the largest amount in the banded cricket, similar to the research of Zielińska et al. [50] and Ribeiro et al. [52]. Zinc affects the structure and operation of several hundred enzymes, participates in the transformation of proteins, fats, and carbohydrates, is essential in the processes of cell growth, division, and death, affects the stabilization and expression of genes, and participates in DNA repair. In our study, the mealworm was also highest in zinc, followed by iron. In the research by Jajić et al. [54], this trace element was found in the highest amount. Insects are a very good source of iron. Moreover, 100 g of dried cricket or mealworm has been shown to provide the recommended daily intake of zinc, copper, and phosphorus [91]. The data showed that crickets and mealworms contain significantly high chemically available Ca, Cu, Mg, Mn, and Zn. Commonly consumed insect species could be excellent sources of bioavailable iron and could provide the platform for an alternative strategy for increased mineral intake in the diets of humans [92].

Minerals are important components of thousands of enzymes and other compounds in the body. Although the content of minerals in the body is small, their importance is huge. Edible insects can provide many of these minerals in abundance and are, therefore, considered as a good source of them. However, the mineral content is highly dependent on the feeding and breeding conditions of the insects. Adamkova et al. [93] found a statistically significant dependence on the type of feed and rearing temperature in the content of minerals, especially Cd and Mg. On the other hand, the concentrations of Na, P, Ca, and Cu in the entire observed range of the feeding or rearing temperature changes were independent of the variables. With properly selected rearing and feeding temperatures, it is possible to obtain mealworms with specific nutritional properties for the target group of consumers [93].

In our research, we also focused on analyzing the content of heavy metals, which are a significant chemical threat to consumer health. Heavy metals are naturally occurring elements in the environment, but their amounts can also enter the environment as a result of human activity. Examples of toxic metals and metalloids are cadmium, mercury, lead, arsenic, chromium, cobalt, nickel, and copper. The mechanism of their toxicity results from interfering with the structure of the body’s cells, e.g., replacing metal ions in proteins or inducing oxidative stress. Heavy metals are not metabolized and must be excreted from the body [94].

Although the insects analyzed in our study were free of some heavy metals, it should be borne in mind that cases of contamination can occur and are documented in the scientific literature. The presence of heavy metals in insects depends to a large extent on the type of feeding and breeding conditions. For example, Jamil and Hussain [95] observed high levels of cadmium and zinc (compared to lower levels of manganese and mercury) in larvae that for one week fed on leaves that had previously been in contact with various metals. Zhang et al. [96] showed the bioaccumulation of cadmium and lead in three out of four herbivorous insects. Additionally, they detected bioaccumulation of mercury. However, the degree of bioaccumulation varied depending on the metal tested, as well as the insect species. The accumulation of heavy metals by mealworms from the soil was described by Vijver et al. [97].

Bednarska et al. [98] described the possibility of bioaccumulation of heavy metals in Cuban cricket, which were exposed to diets enriched with zinc or cadmium. In the case of zinc, accumulation occurred only at the highest level of exposure, while for cadmium, even the lowest level of exposure of crickets to this element caused an increase in the concentration. It has been shown that cadmium concentrations were significantly higher than those allowed by Regulation (EU) 1881/2006 [99] and Regulation (EU) 1275/2013 [100]. Due to the ability to accumulate heavy metals, insects can be used as bioindicators of environmental pollution [101]. From the food and pet food production point of view, it is obvious that the content of heavy metals should be monitored. This phenomenon can have a negative effect not only on the larvae but also on consumers, as there is a risk of consuming the larvae containing heavy metals [12].

Despite the high nutritional value of insects, which can be an excellent source of animal protein in the diet of both humans and companion animals, especially in the case of the latter, there is a lack of research on the safety of using insects in pet food. In a study by Kilburn et al. [102], the apparent digestibility and possible health effects of diets containing varying levels of banded cricket meal administered to healthy adult dogs were evaluated. The apparent digestibility of each nutrient decreased with increasing levels of cricket meal incorporation. However, importantly, all blood count parameters remained within acceptable reference ranges, indicating healthy dogs. Small fluctuations in blood urea nitrogen and hemoglobin levels were observed but not considered significant. Diets containing cricket meal remained highly digestible with over 80% total apparent digestibility. Therefore, crickets have been proven to be an acceptable ingredient in the diet of dogs. Gugołek et al. [54] showed that a diet of mealworm larvae improved the digestibility of acid detergent fiber (ADF). And its effect on the mammalian organism contributes to shifting the place of bacterial fermentation from the cecum to the colon, thus enabling the extraction of additional energy from less digestible components of the diet.

## 5. Conclusions

Insects have gained popularity in recent years due to the fact that they are a more ecological source of animal protein, compared to slaughtered animals such as beef, poultry, or pork. They use less water, emit less greenhouse gases, need much less space for breeding, and can be fed on waste. Edible insects are highly nutritious, but their nutritional composition varies widely. Due to the high protein content, our research confirmed that insects can be a valuable alternative to conventional slaughtered animals in this respect. In addition to protein, some insects can provide valuable fat, rich in polyunsaturated fatty acids, including EPA and DHA, essential for nutrition. Our study, while providing up-to-date knowledge on the nutritional value of insects, also has its limitations. First of all, only two species of insects were analyzed; further research should focus on expanding this number. What is more, since we analyzed fatty acids, the level of amino acids should also be analyzed in future research.

## Figures and Tables

**Table 1 foods-12-04174-t001:** Proximate composition (g/100 g DM) of selected insect species.

Item	Banded Cricket	Mealworm
Dry matter (g/100 g)	98.05 ^a^ ± 0.04	95.56 ^b^ ± 0.56
Crude protein (4.76 × N)	49.03 ^b^ ± 0.08	54.16 ^a^ ± 0.32
Crude protein (5.60 × N)	57.68 ^b^ ± 0.09	63.72 ^a^ ± 0.38
Crude protein (6.25 × N)	65.66 ^b^ ± 0.10	74.41 ^a^ ± 0.42
Crude fat	21.51 ^a^ ± 0.14	9.27 ^b^ ± 0.11
Crude fiber	7.36 ^b^ ± 0.25	8.09 ^a^ ± 0.29
Crude ash	4.84 ^b^ ± 0.27	7.60 ^a^ ± 0.25
Total carbohydrates	0.62 ^a^ ± 0.43	0.63 ^a^ ± 0.83

Means with the same letter in the superscript (a, b) did not differ statistically at *p* = 0.05 (for all lines separately); DM, dry matter; values are expressed as mean (*n* = 3).

**Table 2 foods-12-04174-t002:** Macromineral content (g/100 g DM) of selected insect species.

Item	Banded Cricket	Mealworm
Calcium (Ca)	0.14 ^a^ ± 0.00	0.24 ^b^± 0.02
Phosphorus (P)	0.83 ^a^ ± 0.12	1.24 ^b^ ± 0.09
Potassium (K)	1.10 ^a^ ± 0.33	1.54 ^b^ ± 0.19
Sodium (Na)	0.42 ^b^ ± 0.01	0.37 ^a^ ± 0.02
Magnesium (Mg)	1.06 ^a^ ± 0.01	8.75 ^b^ ± 0.07

Means with the same letter in the superscript (a, b) did not differ statistically at *p* = 0.05 (for all lines separately); DM, dry matter; values are expressed as mean (*n* = 3).

**Table 3 foods-12-04174-t003:** Trace element content (mg/100 g DM) of selected insect species.

Item	Banded Cricket	Mealworm
Copper (Cu)	3.23 ^a^ ± 0.01	2.05 ^b^ ± 0.01
Iron (Fe)	4.21 ^b^ ± 0.04	4.64 ^a^ ± 0.05
Manganese (Mn)	1.89 ^a^ ± 0.01	0.87 ^b^ ± 0.02
Zinc (Zn)	10.03 ^a^ ± 0.04	8.20 ^b^ ± 0.03
Molybdenum (Mo)	0.09 ^b^ ± 0.00	0.16 ^a^ ± 0.00

Means with the same letter in the superscript (a, b) did not differ statistically at *p* = 0.05 (for all lines separately); DM, dry matter; values are expressed as mean (*n* = 3).

**Table 4 foods-12-04174-t004:** Fatty acid profile of BC and ML lipids.

Fatty Acid	BC		ML	
	mg/g of Fat	% of Total Sum of Fatty Acids	mg/g of Fat	% of Total Sum of Fatty Acids
C12:0	0.39 ^a^ ± 0.15	0.06 ^A^ ± 0.04	0.23 ^a^ ± 0.12	0.03 ^A^ ± 0.01
C14:0	3.56 ^a^ ± 0.007	0.54 ^A^ ± 0.05	11.87 ^b^ ± 0.005	1.50 ^B^ ± 0.01
C16:0	153.06 ^b^ ± 0.00	23.38 ^B^ ± 0.02	122.96 ^a^ ± 0.060	15.53 ^A^ ± 0.02
C16:1n-9	2.22 ^a^ ± 0.005	0.34 ^A^ ± 0.18	3.61 ^b^ ± 0.023	0.46 ^A^ ± 0.03
C16:1 n-7	2.68 ^a^ ± 0.007	0.41 ^A^ ± 0.04	7.51 ^b^ ± 0.046	0.95 ^B^ ± 0.01
C17:0	1.33 ^a^ ± 0.008	0.20 ^A^ ± 0.09	4.52 ^b^ ± 0.067	0.57 ^B^ ± 0.05
C18:0	101.42 ^b^ ± 0.020	15.49 ^B^ ± 0.15	52.36 ^a^ ± 0.021	6.61 ^A^ ± 0.00
C18:1 n-9	157.54 ^a^ ± 0.008	24.06 ^A^ ± 0.23	303.52 ^b^ ± 0.009	38.33 ^B^ ± 0.02
C18:1 n-7	3.98 ^b^ ± 0.019	0.61 ^B^ ± 0.06	2.43 ^a^ ± 0.012	0.31 ^A^ ± 0.00
C18:2 n-6	200.59 ^a^ ± 0.060	30.63 ^A^ ± 0.25	269.77 ^b^ ± 0.017	34.06 ^B^ ± 0.03
C20:0	2.85 ^a^ ± 0.074	0.44 ^A^ ± 0.04	4.20 ^b^ ± 0.032	0.53 ^B^ ± 0.01
C18:3 n-3	15.84 ^b^ ± 0.055	2.42 ^B^ ± 0.23	8.15 ^a^ ± 0.006	1.03 ^A^ ± 0.01
C20:1 n-9	0.97 ^a^ ± 0.002	0.15 ^A^ ± 0.07	1.04 ^b^ ± 0.002	0.13 ^A^ ± 0.00
C20:5 n-3	1.94 ± 0.009	0.30 ± 0.01	ND	ND
C24:1 n-9	3.73 ± 0.023	0.57 ± 0.03	ND	ND
C22:6 n-3	3.09 ± 0.015	0.47 ± 0.05	ND	ND
SFA	262.61 ^b^ ± 0.005	40.11 ^B^ ± 0.05	196.14 ^a^ ± 0.015	24.77 ^A^ ± 0.02
MUFA	171.12 ^a^ ± 0.025	26.13 ^A^ ± 0.02	318.10 ^b^ ± 0.026	40.17 ^B^ ± 0.01
PUFA	221.46 ^a^ ± 0.024	33.82 ^A^ ± 0.03	277.92 ^b^ ± 0.017	35.09 ^B^ ± 0.01
Total FA	655.15 ^a^ ± 0.002		792.17 ^b^ ± 0.001	
n-6/n-3	17.70 ^a^ ± 0.08		33.10 ^b^ ± 0.09	
IA	0.43 ^a^ ± 0.02		0.29 ^b^ ± 0.03	
IT	1.04 ^a^ ± 0.01		4.65 ^b^ ± 0.04	

Means with the same letter in the superscript (a, b for mg/g of fat column and A, B for % of total fatty acids) did not differ statistically at *p* = 0.05 (for all lines separately); SFA—saturated fatty acids; MUFA—monounsaturated fatty acids; PUFA—polyunsaturated fatty acids; FA—fatty acids; IA—Index of Atherogenicity; IT—Index of Thrombogenicity; ND—not detected; values are expressed as mean (*n* = 3).

## Data Availability

The data used to support the findings of this study can be made available by the corresponding author upon request.

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
