# Peer review of "Nutritional Value of Banded Cricket and Mealworm Larvae"

_foods, 2023, doi:10.3390/foods12224174_

Round 1
Reviewer 1 Report
Comments and Suggestions for Authors
1. Please mention how novel this study is in the introduction. Why were these two insects chosen? This was only an analysis of a commercial insect that has been well researched. So, how original is it?
2. Line 17-30. To highlight the finding, numerical values are required in the abstract. In addition, some conclusion and applicability of this study should be provided in the final section of the abstract.
3. Line 120. Why was a conversion factor of 6.25 used? A conversion factor of 5.6 or other values can be employed to avoid overestimation of protein content due to the presence of certain amounts of non-protein nitrogen in insects. Please refer to this paper or other publications for more information. Chinarak, K., Panpipat, W., Summpunn, P., Panya, A., Phonsatta, N., Cheong, L. Z., & Chaijan, M. (2021). Insights into the effects of dietary supplements on the nutritional composition and growth performance of sago palm weevil (Rhynchophorus ferrugineus) larvae. Food Chemistry, 363, 130279.
4. Line 126. Please carefully check how to calculate carbohydrate by difference. Fiber is a form of carbohydrate.
5. Line 164-167. A number of replications are required for the analysis.
6. Tables 1-3. Please provide the mean ± standard deviation (SD).
7. Tables 2 and 3 should be consistent. Table 3 requires the full name of minerals.
8. Table 4. Please double-check the significance figure for both mean and standard deviation. There is no need to offer three decimals.
9. The index of atherogenicity (IA) and index of thrombogenicity (IT) can be calculated using the data from Table 4.
10. This study does not provide amino acid compositions. Why? One of the components that determines the nutritional value of an insect is amino acid. The essential amino acid index (EAAI) and biological value (BV) should be provided along with the amino acid profile.
11. What about the cholesterol content of these insects?
12. “The BC lipids also contained EPA and DHA, which were not found in the ML lipids.” Please make a discussion on this data.
13. A discussion on the effects of different feed compositions on the chemical composition of both insects is required.
Comments on the Quality of English Language
Minor editing of English language required.
Author Response
Response to Reviewer 1 comments
Point 1: Please mention how novel this study is in the introduction. Why were these two insects chosen? This was only an analysis of a commercial insect that has been well researched. So, how original is it?
Response: This study contributes new knowledge on the nutritional value of edible insects. In this research we included three different nitrogen conversion factors for crude protein content. Our results partially confirm previous studies by other authors, although they provide new information on the content of fatty acids.
Point 2: Line 17-30. To highlight the finding, numerical values are required in the abstract. In addition, some conclusion and applicability of this study should be provided in the final section of the abstract.
Response: We improved and added numerical values in the abstract.
Point 3: Line 120. Why was a conversion factor of 6.25 used? A conversion factor of 5.6 or other values can be employed to avoid overestimation of protein content due to the presence of certain amounts of non-protein nitrogen in insects. Please refer to this paper or other publications for more information. Chinarak, K., Panpipat, W., Summpunn, P., Panya, A., Phonsatta, N., Cheong, L. Z., & Chaijan, M. (2021). Insights into the effects of dietary supplements on the nutritional composition and growth performance of sago palm weevil (Rhynchophorus ferrugineus) larvae. Food Chemistry, 363, 130279.
Response: Of course, we agree with this suggestion, currently it is emphasized that 6.25 should not be used, in accordance with what we have established, in other recent works the authors provide different nitrogen conversion factors to protein. In our research, we took into account three nitrogen conversion factors - 4.76, 5.60 and 6.25. The discussion and results in Table 1 have been supplemented in this respect.
Point 4: Line 126. Please carefully check how to calculate carbohydrate by difference. Fiber is a form of carbohydrate.
Response: Thank you for this comment, however, in the case of determining the proximate composition calculating the carbohydrate content from the difference is a standard operation. Total carbohydrates (TC) - nitrogen-free extracts (NFE) were calculated taking into account crude fiber (not dietary fiber) in accordance with the commonly accepted formula
Point 5: Line 164-167. A number of replications are required for the analysis.
Response: Thank you for this suggestion, we have added information about the number of repetitions.
Point 6: Tables 1-3. Please provide the mean ± standard deviation (SD).
Response: We provided SD.
Point 7: Tables 2 and 3 should be consistent. Table 3 requires the full name of minerals.
Response: We added the full names of minerals, however we decided to leave the Tables separated, because macrominerals and trace elements are expressed in different units.
Point 8: Table 4. Please double-check the significance figure for both mean and standard deviation. There is no need to offer three decimals.
Response: Thank you for your suggestion. We agree with the opinion. The precision of the method given in Table 4 is the standard deviation of the mean (SD) for 3 repetitions. In the study we corrected the calculations by rounding the values to two decimal places.
Point 9: The index of atherogenicity (IA) and index of thrombogenicity (IT) can be calculated using the data from Table 4.
Response: Thank you for your comment. Health indices were calculated and added to manuscript.
The indices of atherogenicity (IA) and thrombogenicity (IT) were calculated according to the formulas:
IA = [C12 : 0 + (4 × C14 : 0) + C16 : 0]/ΣUFA
IT = (C14 : 0 + C16 : 0 + C18 : 0)/[(0.5 × ΣMUFA) + (0.5 × Σn − 6 PUFA) + (3 × Σn − 3 PUFA) + (n − 3 / n − 6)]
Point 10: This study does not provide amino acid compositions. Why? One of the components that determines the nutritional value of an insect is amino acid. The essential amino acid index (EAAI) and biological value (BV) should be provided along with the amino acid profile.
Response: Of course, the quality of protein estimated based on the composition of amino acids and calculated indicators of the nutritional value of protein, such as the EAA index, or indicators estimated based on biological methods including biological value (BV) or protein efficiency ratio (PER) and other commonly used ones, is very important, but in this manuscript we decided to focus on fat and its quality, i.e. fatty acid profile.
Point 11: What about the cholesterol content of these insects?
Response: We did not analyze the cholesterol content in our research, therefore we do not discuss it.
Point 12: “The BC lipids also contained EPA and DHA, which were not found in the ML lipids.” Please make a discussion on this data.
Response: As suggested by the reviewer, a discussion has been expanded in this field (lines 403-416).
Point 13: A discussion on the effects of different feed compositions on the chemical composition of both insects is required.
Response: We added information about the effects of different feed compositions on the chemical composition of insects (lines 73-90).
Reviewer 2 Report
Comments and Suggestions for Authors
Dear Authors,
Your manuscript entitled: “Nutritional value of banded cricket and mealworm larvae” has been sent for my consideration. The data are certainly of great interest, as insects are becoming increasingly important for the food/feed sector. The manuscript is well written and covers important aspects of insects for the feed and food sector.
Reading through the manuscript, some things caught my attention, and I'd appreciate any comments and modifications you can suggest.
Line 35: Reference should be made to the WHO reports in this case.
Line 43: The FAO report has made a significant contribution to raising awareness of insects as a potential food resource in the western world, but since 2013 more has definitely been done in terms of insect evaluation, cultivation regulations and the legal situation. Therefore, further references on country-specific or continent-specific insects (EFSA, IPIFF, ec. pp) should be provided here.
Line 96-98: I would suggest not citing Walker and Linnaeus unless you actually used the old strains in your research.
Line 103-105: What were the insects fed? Did the feeding material undergo an analysis to determine its composition (macronutrients/micronutrients)? Could you please provide more specific information about the insect larvae's age?
Line 124-127: Besides fats, proteins, minerals, and carbohydrates, are other minor components of the insect taken into account in this calculation?
Tab. 1-3: The standard errors should be entered into the table
Tab. 4: Are the error deviations standard error of the mean (SEM) or given as standard deviation (SD)?
Line 190: Within Tabl. 4 the letter “n” is already dedicated to the labeling of carbon-carbon double bonds position.
Line 415: What heavy metals were examined by you and how can you exclude that others were not present?
As EFSA's safety assessment is especially concerned with dried insect larvae, I think the oxidative status of both insect larvae should be valued (peroxide value and/or p-anisidine value)
The manuscript would be very welcome after revision - thank you very much!
Comments on the Quality of English Language
xxx
Author Response
Response to Reviewer 2 comments
Reviewer: Dear Authors, Your manuscript entitled: “Nutritional value of banded cricket and mealworm larvae” has been sent for my consideration. The data are certainly of great interest, as insects are becoming increasingly important for the food/feed sector. The manuscript is well written and covers important aspects of insects for the feed and food sector. Reading through the manuscript, some things caught my attention, and I'd appreciate any comments and modifications you can suggest.
Response: Thank you very much for many valuable suggestions. We have tried to improve the manuscript taking into account the reviews.
Point 1: Line 35: Reference should be made to the WHO reports in this case.
Response: We added a reference to FAO statistics ([1]).
Point 2: Line 43: The FAO report has made a significant contribution to raising awareness of insects as a potential food resource in the western world, but since 2013 more has definitely been done in terms of insect evaluation, cultivation regulations and the legal situation. Therefore, further references on country-specific or continent-specific insects (EFSA, IPIFF, ec. pp) should be provided here.
Response: We added information about continent-specific insects (lines 52-60).
Point 3: Line 96-98: I would suggest not citing Walker and Linnaeus unless you actually used the old strains in your research.
Response:
Thank you, we fully agree, these terms have been removed.
Point 4: Line 103-105: What were the insects fed? Did the feeding material undergo an analysis to determine its composition (macronutrients/micronutrients)? Could you please provide more specific information about the insect larvae's age?
Response: We did not have feed material to perform analyses, but we were assured by the insect breeder from whom we purchased the insects that they were fed with complete feed standard for edible insects. The insects were in the subimago stage.
Point 5: Line 124-127: Besides fats, proteins, minerals, and carbohydrates, are other minor components of the insect taken into account in this calculation?
Response: In this case, unfortunately not.
Point 6: Tab. 1-3: The standard errors should be entered into the table
Response: We added means and standard deviations (SD).
Point 7: Tab. 4: Are the error deviations standard error of the mean (SEM) or given as standard deviation (SD)?
Response: Thank you very much for pointing this out. The precision of the method given in Table 4 is the standard deviation of the mean (SD) for three repetitions. In the study we corrected the calculations by rounding the values to two decimal places.
Point 8: Line 190: Within Tabl. 4 the letter “n” is already dedicated to the labeling of carbon-carbon double bonds position.
Response: Thank you for suggestion. The letter "n" connected to the number of double bonds is included in the table and is a standard chemical designation of the fatty acid family, which is also repeated in the text of the publication. The symbol omega can be used interchangeably, but in the vast majority of studies it is given by the formula (n - x) where n is the number of carbons in the chain and x ( x = 3; n = 6; x = 9) is the first double bond counting the carbons starting from carboxyl group of the fatty acid and not from the CH3 group, i.e. omega. The letter denoting the number of parallel repetitions, taken as n, appears only in the table description and is an element of statistical information. In response to the comment, it was decided to correct the letter "n" with the letter "n" written in italics.
Point 9: Line 415: What heavy metals were examined by you and how can you exclude that others were not present?
Response: Of course, we do not rule out that there were no other heavy metals. We analyzed the content of Cd, Co, Cr, Ni, Pb and the statement about the absence of heavy metals applies only to the analyzed elements.
Point 10: As EFSA's safety assessment is especially concerned with dried insect larvae, I think the oxidative status of both insect larvae should be valued (peroxide value and/or p-anisidine value).
Response: We fully agree with this and we are considering conducting separate studies in this respect.
Point 11: The manuscript would be very welcome after revision - thank you very much!
Response: The manuscript has been improved in accordance with comments. Thank you for many valuable suggestions!
Round 2
Reviewer 1 Report
Comments and Suggestions for Authors
All points raised by reviewers were addressed and answered point-by-point. So, it can be accepted as is.
Author Response
Response to Reviewer 1 comments
Reviewer: All points raised by reviewers were addressed and answered point-by-point. So, it can be accepted as is.
Response: Dear Reviewer, we would like to thank you very much for the entire review process and suggestions, which significantly contributed to increasing the scientific value of our manuscript.
Reviewer 2 Report
Comments and Suggestions for Authors
Dear Authors,
Your revised manuscript entitled “Nutritional value of banded cricket and mealworm larvae” has been sent for another review.
Thanks for reading my comments and updating the manuscript. There are still a few things open and I'd like to mention them briefly:
· Line 133- 135: It is unfortunate that you do not have any information about the exact feed batch. It's still important for the reader to get an idea of what the "standard feed" is so they can categorize and compare the data. Were both insects fed with the same feed? In my opinion, this is important to mention, as the nutrient composition of the insects is statistically analysed later in the results section
· Line 157-160: Was this equation already been used by others and if so, please cite the reference?
· Line 496: For evaluating the statement “Although the insects analyzed in our study were free of some heavy metals”, the analytical parameters LOD and LOQ must be specified in the material and methods section since they provide information about analytical sensitivity.
Thank you for taking the time to respond to my comments
Author Response
Response to Reviewer 2 comments
Reviewer: Dear Authors, your revised manuscript entitled “Nutritional value of banded cricket and mealworm larvae” has been sent for another review. Thanks for reading my comments and updating the manuscript. There are still a few things open and I'd like to mention them briefly. Thank you for taking the time to respond to my comments.
Response: Dear Reviewer, we would like to thank you very much for the entire review process and suggestions, which significantly contributed to increasing the scientific value of our manuscript.
Point 1: Line 133- 135: It is unfortunate that you do not have any information about the exact feed batch. It's still important for the reader to get an idea of what the "standard feed" is so they can categorize and compare the data. Were both insects fed with the same feed? In my opinion, this is important to mention, as the nutrient composition of the insects is statistically analysed later in the results section
Response: We fully agree with this comment, however, the exact component composition of insect feed is the trade secret of the breeder. We have access to the analytical composition of the feed only (crude protein 16%, crude fat 3%, crude ash 8%, crude fiber 6%, vitamin A 8500 IU/kg, vitamin D3 1100 IU/kg, vitamin E 60 mg/kg). Both insect species were fed the same feed, we added this information (lines 132-134).
Point 2: Line 157-160: Was this equation already been used by others and if so, please cite the reference?
Response: We added a reference. This equation is also used in relation to the proximate composition of insects, and other components and raw materials. Below are some examples of publications:
- Chakravorty, J.; Ghosh, S.; Megu, K..; Jung, C..; Meyer-Rochow, V.B. Nutritional and anti-nutritional composition of Oecophylla smaragdina (Hymenoptera: Formicidae) and Odontotermes sp. (Isoptera: Termitidae): Two preferred edible insects of Arunachal Pradesh, India. J Asia-Pacif Entomol 2016, 19, 711-720, doi: 10.1016/j.aspen.2016.07.001.
- Akinjogunla, V.F.; Lawal-Are, A.O.; Soyinka, O.O. Proximate composition and mineral contents of Mangrove Oyster (Crassostreagasar) from Lagos Lagoon, Lagos, Nigeria. Nig J Fish Aquac 2017, 5(2), 36-49.
- Grabowski, N.; Ty, Ch.; Keo, S.; Lertpatarakomol, R.; Kajaysri, J.; Kang, K.; Miech, P.; Ploetz, M.; Mitchaothai, J. Proximate composition of thai and cambodian ready-to-eat insects. J Food Qual 2021, 2021, 1-6, doi:10.1155/2021/9731464.
- Son, Y.J.; Hwang, I.K.; Nho, C.W.; Kim, S.M.; Kim, S.H. Determination of carbohydrate composition in mealworm (Tenebrio molitor L.) larvae and characterization of mealworm chitin and chitosan. Foods 2021, 10(3), 640, doi:10.3390/foods10030640.
- Jajić, I.; Krstović, S.; Petrović, M.; Urošević, M.; Glamočić, D.; Samardžić, M.; Popović, A.; Guljaš, D. Changes in the chemical composition of the yellow mealworm (Tenebrio molitor L.) reared on different feedstuffs. J Anim Feed Sci 2022, 31(2), 191-200, doi:10.22358/jafs/147848/2022.
- Zielińska, E.; Baraniak, B.; Karaś, M.; Rybczyńska, K.; Jakubczyk, A. Selected species of edible insects as a source of nutrient composition. Food Res Int 2015, 77(3), 460-466, doi:10.1016/j.foodres.2015.09.008.
- Aletan, U.; Kwazo, H. Analysis of the proximate composition, anti-nutrients and mineral content of Maerua Crassifolia leaves. Nig J Basic Appl Sci 2020, 27, 89-96, doi:10.4314/njbas.v27i1.12.
Point 3: Line 496: For evaluating the statement “Although the insects analyzed in our study were free of some heavy metals”, the analytical parameters LOD and LOQ must be specified in the material and methods section since they provide information about analytical sensitivity.
Response: The material and methods chapter has been supplemented (lines 182-184), information in the results section has been corrected (lines 249-250). In the supplementary materials we have added a table showing the mineral composition of certified reference material (Table S1, page 17).